# Induced abortion in Africa: A systematic review and meta-analysis

**Teklehaimanot Gereziher Haile**[1] *, **Teklehaymanot Huluf Abraha**[2], **Gebreamlak Gebremedhn Gebremeskel**[3], **Kidane Zereabruk**[3], **Tesfay Hailu Welu**[4], **Teklit Grum**[2], **Negasi Asres**[5]

1 Department of Maternity and Neonatal Nursing, School of Nursing, College of Health Sciences and Comprehensive Specialized Hospital, Aksum University, Aksum, Tigray, Ethiopia, 2 Department of Reproductive and Family Health, School of Public Health, College of Health Sciences and Comprehensive Specialized Hospital, Aksum University, Aksum, Tigray, Ethiopia, 3 Department of Adult Health Nursing, School of Nursing, College of Health Sciences and Comprehensive Specialized Hospital, Aksum University, Aksum, Tigray, Ethiopia, 4 Department of Midwifery, College of Health Sciences and Comprehensive Specialized Hospital, Aksum University, Aksum, Tigray, Ethiopia, 5 Department of Epidemiology and Biostatistics, School of Public Health, College of Health Sciences and Comprehensive Specialized Hospital, Aksum University, Aksum, Tigray, Ethiopia

* teklehaimg@gmail.com

## Abstract

### Background

One of the main factors contributing to maternal morbidity and mortality is induced abortion. The WHO estimates that over 44 million induced abortions take place annually around the world. The majority of these abortions—about 50%—are unsafe, significantly increasing maternal morbidity and contributing to 13% of maternal deaths. Thus, this review aimed to estimate the pooled prevalence of induced abortion and its associated factors in Africa.

### Methods

To find literature on the prevalence of induced abortion and its associated factors, a thorough search of the internet databases such as PubMed/MEDLINE, African Journals Online, and Google Scholar was conducted. The data were extracted using a structured method of data collection. Software called STATA 14 was used to do the analysis. funnel plot and Egger regression test were used to evaluate potential publication bias. $I^2$ statistics and Cochrane's $Q$ were used to measure the heterogeneity at a p-value < 0.05.

### Results

976 studies were found through a thorough search of electronic databases. Finally, 46 full-text abstract papers were included in this study. The estimated pooled prevalence of induced abortion was 16% (95% CI: 13%-19%). According to the sub-group analysis, most studies were conducted in Ethiopia, and the pooled prevalence was 19% (95% CI: 10%–30%). Similarly, the subgroup analysis by year of study showed that the prevalence of induced abortion was 39% (95% CI: 17%–64%) among studies conducted in 2019.

**Data Availability Statement:** All relevant data are within the paper and its supporting information files.

**Funding:** The author(s) received no specific funding for this work.

**Competing interests:** The authors have declared that no competing interests exist.

**Abbreviations:** AOR, Adjusted Odds Ratio; CI, Confidence Interval; PRISMA, Preferred Reporting Items for Systematic reviews and Meta-Analysis; WHO, World Health Organization.

## Conclusion

The results of this study thus imply that the pooled prevalence of induced abortion is higher than that of earlier studies that were published in some nations. the data from this study are needed to support reproductive and adolescent health programmers and policymakers and to formulate recommendations for future clinical practice and guidelines.

## Introduction

Induced abortion is defined as the intentional termination of pregnancy before the 28th week of gestation or before the fetus is born with a weight less than 1000 grams that (i.e., for developing nations) cannot live independently outside of the womb. The World Health Organization (WHO) classifies induced abortion as safe abortion," which is done by a trained health provider and at an appropriate gestational age with services recommended by the WHO [1].

The frequency of induced abortions varies widely. In Western Europe, it was roughly 12 per 1000 women between the ages of 15 and 44, but it was 43 in Eastern Europe [2]. The WHO estimates that over 44 million induced abortions take place annually around the world. The majority of these abortions—about 50%—are unsafe, significantly increasing maternal morbidity and contributing to 13% of maternal deaths [3, 4]. Evidence suggests that induced abortions are more prevalent in nations where abortion is prohibited or regulated than in those where it is permitted [2].

One of the main factors contributing to maternal morbidity and mortality is induced abortion [5]. Six out of ten unplanned pregnancies and three out of ten pregnancies worldwide resulted in an induced abortion [6]. 97% of unsafe abortions take place in developing nations, accounting for around 45% of all abortions [7]. In sub-Saharan Africa, there are approximately 33 unsafe abortions per 1,000 women aged 15 to 49 performed each year [8]. Abortion remains one of the most sensitive sexual and reproductive behaviors because of social stigma, privacy concerns, and the fear of legal sanctions [9].

According to the Human Reproduction Program, there were an estimated 121 million unintended pregnancies in women aged 15–49 years each year between 2015 and 2019, and 61% of these ended in abortion. Moreover, 45% of these abortions were unsafe in 2017 [10]. The poorest women with the fewest resources are the most likely to experience complications from unsafe abortions. Maternal death due to complications of an unsafe abortion was the major cause of hospital admission. Each year, 4.7–13.2% of maternal deaths can be attributed to unsafe abortion [5]. In developing regions, that number rises to 220 deaths per 100,000 unsafe abortions [5, 7, 11].

Induced abortion remains a major public health problem confronting African women [7]. The specific Sustainable Development Goals (SDGs) known as the health goal (goal No. 3) aim to ensure healthy lives and promote well-being for all at all ages, with one of the important targets being to ensure universal access to reproductive health care services, including family planning, information, and education, and the integration of reproductive health into national strategies and programs [12].

To the best of our knowledge, this is the first systematic review and meta-analysis of the prevalence of induced abortion and its associated factors in Africa. Therefore, data on systematic reviews and meta-analyses of induced abortion are needed to support reproductive and adolescent health programmers and policymakers and to formulate recommendations for future clinical practice and guidelines.

## Materials and methods

### Study design, search strategy and data source

A systematic review and meta-analysis were done using published and unpublished articles on the prevalence of induced abortion and its associated factors in Africa. To find literature on the prevalence of induced abortion and its associated factors, a thorough search of the internet databases such as PubMed/MEDLINE, African Journals Online, and Google Scholar was conducted. Additionally, references in studies that passed screening were checked. Since there is no article published before 1999, we included all articles published from March 1999 to May 1, 2023, in this study.

The following keywords were used in the search: "induced abortion", "abortion", "termination of pregnancy", "prevalence", "incidence", "proportion", "determinants", "criminal abortion", "reproductive age women", "associated factors", "risk factors", "magnitude", and "names of each African nation." The Boolean operators "AND" and "OR" were combined as necessary, and two authors (TGH and THA) independently conducted the search. According to the recommended reporting items for systematic review and meta-analysis (PRISMA) standard, this systematic review and meta-analysis were reported [13] (S1 Table).

### Study selection

Studies that were done in each African nation and reported on the prevalence of induced abortion and its associated factors were chosen for the meta-analysis. Duplicate files were removed after exporting all articles read from a few databases to Endnote X8. Three investigators (GGG, TG, and NA) screened the remaining articles and abstracts for inclusion in the full-text appraisal. Before being included in the review, two reviewers (KZ and THW) independently evaluated the publications.

### Data extraction and quality assessment

The data were separately extracted by three authors (TGH, GGG, and TG) using a structured method of data collection. Data extraction format made in a Microsoft Excel spreadsheet. Two reviewers (TGH and THA) separately examined the titles, abstracts of each reference obtained, and the full-text search results to categorize possibly qualifying papers. Title, author name, study design, research type, year of study, research base (including population and hospital-based research), sample size, response rate, study area, study quality score, and prevalence contained were all included in the data extraction.

All studies quality was evaluated using the Newcastle-Ottawa Quality Assessment Scale [14]. Assigning a maximum of 10 stars for the risk of bias in three areas—study group selection (4 or 5 stars), group comparability (2 stars), and ascertainment of the result of interest or exposure (3 stars)—is the main method used to create this scale. According to the aggregate quality score, there is a high, moderate, and low risk of bias for ratings of 0–3, 4–6, and 7–10 stars, respectively (S2 Table). The two reviewers' disagreements were resolved through dialogue and discussion.

### Eligibility criteria

The standards used to choose which research will be included in the analysis are based on eligibility criteria for studies. These standards are crucial for guaranteeing that the chosen research is reliable, legitimate, and of a high enough caliber to yield insightful findings (Table 1).

### Publication bias and heterogeneity

$I^2$ statistics were used to assess the statistical heterogeneity. Heterogeneity was categorized as low, medium, and high heterogeneity, with values of $I^2$ of 25%, 50%, and 75%, respectively

**Table 1. Criteria for considering studies for this study.**

| Eligibility criteria | |
|---|---|
| **Inclusion criteria** | Design: All types of observational study designs |
| | Publication type: Both published and unpublished articles |
| | Population: All reproductive age groups of women |
| | Study Setting: Studies conducted in Africa, which are institutional-based. |
| | Language: Published articles written exclusively in English were considered in this study. |
| | This study was included all published articles from March 1999 to May 1, 2023. |
| | Outcome: Prevalence of induced abortion and associated factors |
| **Exclusion criteria** | Studies that did not reveal the prevalence of induced abortion and its associated factors |
| | Methodological problems (This including: inadequate sample size, sampling techniques, measurement error, incomplete data, inadequate statistical analysis, or any other relevant methodological limitations identified during the review process). |

[15]. If there is heterogeneity between the included articles. The authors, therefore, will use a meta-analysis of random effects to estimate the aggregate pooled prevalence of induced abortion in Africa. Methods like the funnel plot and Egger regression test were used to evaluate potential publication bias. Significant publication bias was defined as the existence of a P-value of 0.05 [16].

## Statistical analysis/ data synthesis and sub-group analyses

Software called STATA 14 was used to do the analysis. The prevalence of induced abortion among reproductive age groups in Africa was displayed using forest plots. $I^2$ statistics and Cochrane's $Q$ were used to measure the heterogeneity, and a p-value of less than 0.05 was used to proclaim it [15]. A funnel plot and Egger's test were used to assess publication bias. Different study characteristics, such as study year and study country, were subjected to subgroup analyses. A p-value of less than 0.05 was used to corroborate the publication bias test using the Egger regression asymmetry [17]. Additionally, the 'trim and fill' method developed by Duval and Tweedie was used to calculate the approximate number of studies that the meta-analysis was missing [18]. Tables and forest plots with 95% confidence intervals (CI) were used to present the results. Due to the significant degree of heterogeneity among the included publications, a random-effects model was adopted [19].

## Data management

Based on the standards for inclusion and exclusion, a predetermined framework was developed to direct the screening and selection procedures. Before starting the data extraction, the tool was tested and updated. The search results were first submitted to EndNote 8x software in order to remove duplicates.

## Data items

The first author, publishing year, nation, sample size, publication type, study area, study design, and response rate were all included in the data extraction.

## Outcomes and prioritization

The primary outcome is the prevalence of induced abortion and its associated factors in Africa.

## Results

### Articles included in the meta-analysis

At first, 976 studies were found through a thorough search of electronic databases. Titles and abstracts were screened, and duplicated or irrelevant articles were removed using EndNote 8x. Accordingly, we eliminated 587 duplicate articles. After reviewing the remaining 389 publications based on their titles and abstracts, we found that 254 articles were disqualified because the studies in those articles didn't match the criteria for inclusion or exclusion. Following a second review of 135 articles, 89 were disregarded because 89 of the studies did not focus on induced abortion (n = 19), lack of results (n = 15), or study areas outside of Africa (n = 55).

Finally, 46 full-text abstract papers were included in this systematic review and meta-analysis based on the pre-defined criteria and quality assessment. However, within these 46 studies, there were four studies that had a case-control study design which designed to investigate associations between exposures and outcomes, rather than directly estimating prevalence. As a result, these four case-control studies were still considered in the analysis of associated factors, as their study design allowed for the examination of potential risk factors or predictors. The remaining 42 studies were suitable for inclusion in the meta-analysis of pooled prevalence. A PRISMA flow chart of the study selection shows the specific steps of the screening procedure (Fig 1).

### Characteristics of the study

This systematic review and meta-analysis study include 46 articles with a combined sample size of 195,660. The smallest sample size was 64, which was drawn from Nigerian research [20]. Whereas, the largest sample size was 146,713, according to a study from the African nation of Congo [21]. Different authors conducted induced abortion studies in Ethiopia in different years, with the highest prevalence (68.7%) and the lowest prevalence (1.1%) [22, 23]. The Newcastle-Ottawa Scale quality assessment criteria for each primary study's quality score indicated no appreciable risk; hence, all the studies were considered in this systematic review and meta-analysis. The detailed characteristics of the included articles are presented in Table 2.

### The pooled prevalence of induced abortion in Africa

In this section of Mata analysis, we included 42 studies, and the estimated pooled prevalence of induced abortion was 16% (95% CI: 13%-19%), but with a significantly high level of heterogeneity among the studies in the random-effects model analysis ($I^2$ = 99.28%, p≤0.000) (Fig 2).

### Publication bias

In this study, to assess and adjust any publication bias, we first applied a funnel plot based on the assumption that the effect sizes of all the studies are normally distributed around the center of a funnel plot in cases where there is no publication bias. When observing asymmetry in the funnel plots, we test using Egger's bias test and apply the trim-and-fill method to first trim the studies that cause asymmetry in the funnel plot so that the overall effect estimate produced by the remaining studies can be considered minimally affected by publication bias, and then to fill the imputed missing studies in the funnel plot based on the bias-corrected overall estimate.

Therefore, the result shown in Fig 3 seems to have an asymmetrical distribution in the funnel plot. But Egger's tests for the small study effect were highly non-significant for the presence of publication bias (p = 0.426). This indicates there was no publication bias among the included studies in estimating the pooled prevalence of induced abortion. Additionally, a trim-and-fill

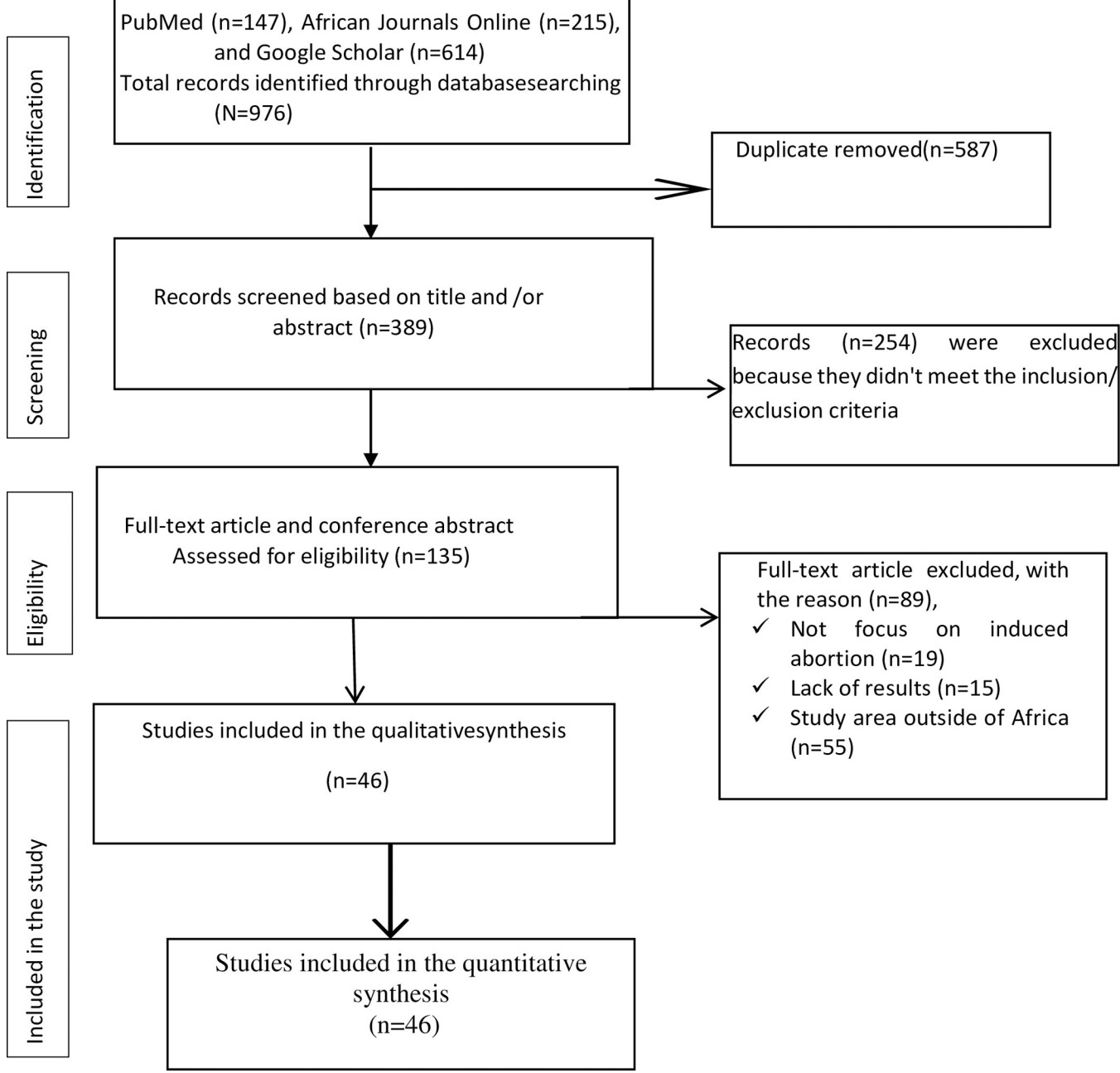

**Fig 1. Selection of studies for a systemic review and meta-analysis for prevalence of induced abortion and its associated factors in Africa.**

analysis for the prevalence of induced abortion was done so as to reduce and correct publication bias in the studies. The result showed no study was imputed for missing studies, and the estimated pooled prevalence was also approximately similar to the unadjusted prevalence.

## Subgroup analysis

In this study, we conducted a subgroup meta-analysis. The studies were grouped based on the study area and study year, and an overall effect size was computed for each group. The aim of

**Table 2. Characteristics of studies considered in this systematic review and meta-analysis of the prevalence of induced abortion in Africa.**

| Authors | Study Year | Study Area | Study Design | Sample Size | Case | Prevalence (%) | Quality score based on NOS |
|---|---|---|---|---|---|---|---|
| Sahile AT et al. [22] | 2019 | Ethiopia | Cross-Sectional | 422 | 290 | 68.7 | 8 |
| Mitiku S et al. [24] | 2015 | Ethiopia | Cross-Sectional | 461 | 27 | 5.9 | 8 |
| Gebeyehu D et al. [25] | 2015 | Ethiopia | RCS | 194 | 69 | 35.6 | 9 |
| Meseret G. et al. [23] | 2013 | Ethiopia | Cross-Sectional | 611 | 7 | 1.1 | 9 |
| Denberu B et al. [26] | 2015 | Ethiopia | Unmatched CC | 110 cases, 220 controls | NM | NM | 7 |
| Abebe M. et al. [27] | 2021 | Ethiopia | Unmatched CC | 103 cases, 309 controls | NM | NM | 8 |
| Megersa et al. [28] | 2017 | Ethiopia | Unmatched CC | 147 cases, 295 controls | NM | NM | 7 |
| Tesfaye G. et al. [29] | 2010 | Ethiopia | Cross-Sectional | 400 | 49 | 12.3 | 7 |
| Tesfaye B et al. [30] | 2017 | Ethiopia | Cross-Sectional | 247 | 73 | 29.6 | 8 |
| Gelaye et al. [31] | 2011 | Ethiopia | Cross-Sectional | 493 | 32 | 6.5 | 8 |
| Megersa A. et al. [32] | 2018 | Ethiopia | Cross-Sectional | 422 | 182 | 43.1 | 8 |
| Bekele D et al. [33] | 2012 | Ethiopia | Cross-Sectional | 340 | 34 | 10 | 7 |
| Senbeto E. et al. [34] | 2003 | Ethiopia | Cross-Sectional | 1346 | 65 | 4.8 | 9 |
| Zeleke AM et al. [35] | 2018 | Ethiopia | Cross-Sectional | 422 | 61 | 14.5 | 8 |
| Jamie H.A. et al. [36] | 2019 | Ethiopia | Cross-Sectional | 611 | 261 | 42.7 | 9 |
| Nigussie et al. [37] | 2018 | Ethiopia | Cross-Sectional | 420 | 79 | 18.8 | 8 |
| Lentiro et al. [38] | 2017 | Ethiopia | Cross-Sectional | 404 | 55 | 13.6 | 8 |
| Bell S.O. et al. [39] | 2020 | Burkina Faso | Cross-Sectional | 1000 | 40 | 4 | 9 |
| Ilboudo et al. [40] | 2012 | Burkina Faso | Cross-Sectional | 304 | 37 | 12 | 7 |
| Fatusi A, et al. [21] | 2016 | Congo | Cross-Sectional | 146,713 | 27,590 | 18.8 | 9 |
| Geelhoed DW. et al. [41] | 1999 | Ghana | Cross-Sectional | 2137 | 482 | 22.6 | 9 |
| Mote C.V. et al. [42] | 2008 | Ghana | Cross-Sectional | 408 | 87 | 21.32 | 8 |
| Klutsey EE et al. [43] | 2012 | Ghana | Unmatched CC | 76 cases, 304 controls | NM | NM | 7 |
| Ahiadeke C. [44] | 1997 | Ghana | Cross-Sectional | 1000 | 17 | 1.7 | 9 |
| Baruwa OJ. et al. [45] | 2017 | Ghana | Cross-Sectional | 18,116 | 3702 | 20.43 | 9 |
| Adjei et al. [46] | 2011 | Ghana | Cross-Sectional | 2723 | 101 | 3.7 | 9 |
| Simmelink AM et al. [47] | 2016–2017 | Kenya | PC | 866 | 103 | 11.9 | 8 |
| Lugaliki AD. [48] | 2013 | Kenya | Cross-Sectional | 329 | 125 | 38 | 7 |
| Mohamed et al. [49] | 2012 | Kenya | Cross-Sectional | 1000 | 48 | 4.8 | 9 |
| Okereke CI. [50] | 2010 | Nigeria | Cross-Sectional | 309 | 62 | 20.2 | 7 |
| Okonofua FE. et al. [51] | 1999 | Nigeria | Cross-Sectional | 176 | 19 | 11 | 7 |
| Obiyan et al. [20] | 2019 | Nigeria | Mixed-Study | 64 | 38 | 59.4 | 7 |
| Murray N. et al. [52] | 2002 | Nigeria | Cross-Sectional | 602 | 247 | 41 | 7 |
| Okonofua F. et al. [53] | 2009 | Nigeria | Cross-Sectional | 1842 | 751 | 40.77 | 9 |
| Bankole A. et al. [54] | 2012 | Nigeria | Cross-Sectional | 1000 | 33 | 3.3 | 9 |
| Ajayi et al. [55] | 2016 | South Africa | Cross-Sectional | 1709 | 325 | 19 | 9 |
| Keogh S.C. et al. [56] | 2013 | Tanzania | Cross-Sectional | 1000 | 36 | 3.6 | 9 |
| Kimbwereza FA et al. [57] | 2019 | Tanzania | Cross-Sectional | 342 | 19 | 5.6 | 7 |
| Mamboleo N. [58] | 2012 | Tanzania | Cross-Sectional | 116 | 26 | 22.4 | 7 |
| Prada E. et al. [59] | 2013 | Uganda | Cross-Sectional | 1000 | 39 | 3.9 | 9 |
| Ndari G. [60] | 2019 | Chad | Cross-Sectional | 384 | 112 | 29.17 | 7 |
| Polis C.B. et al. [61] | 2015 | Malawi | Cross-Sectional | 1000 | 38 | 3.8 | 9 |
| Frederico et al. [62] | 2017 | Mozambique | Cross-Sectional | 1076 | 99 | 9.2 | 9 |
| Bell SO et al. [63] | 2018 | Co^te d'Ivoire | Cross-Sectional | 1000 | 28 | 2.8 | 8 |
| Levandowski BA. et al. [64] | 2009 | Malawi | Cross-Sectional | 1000 | 23 | 2.3 | 9 |
| Dahlbäck E. et al. [65] | 2005 | Zambia | Cross-Sectional | 87 | 34 | 39.1 | 7 |

CC: Case Control, PC: Prospective Cohort, NM: Not Mentioned, NOS: Newcastle Ottawa Scale

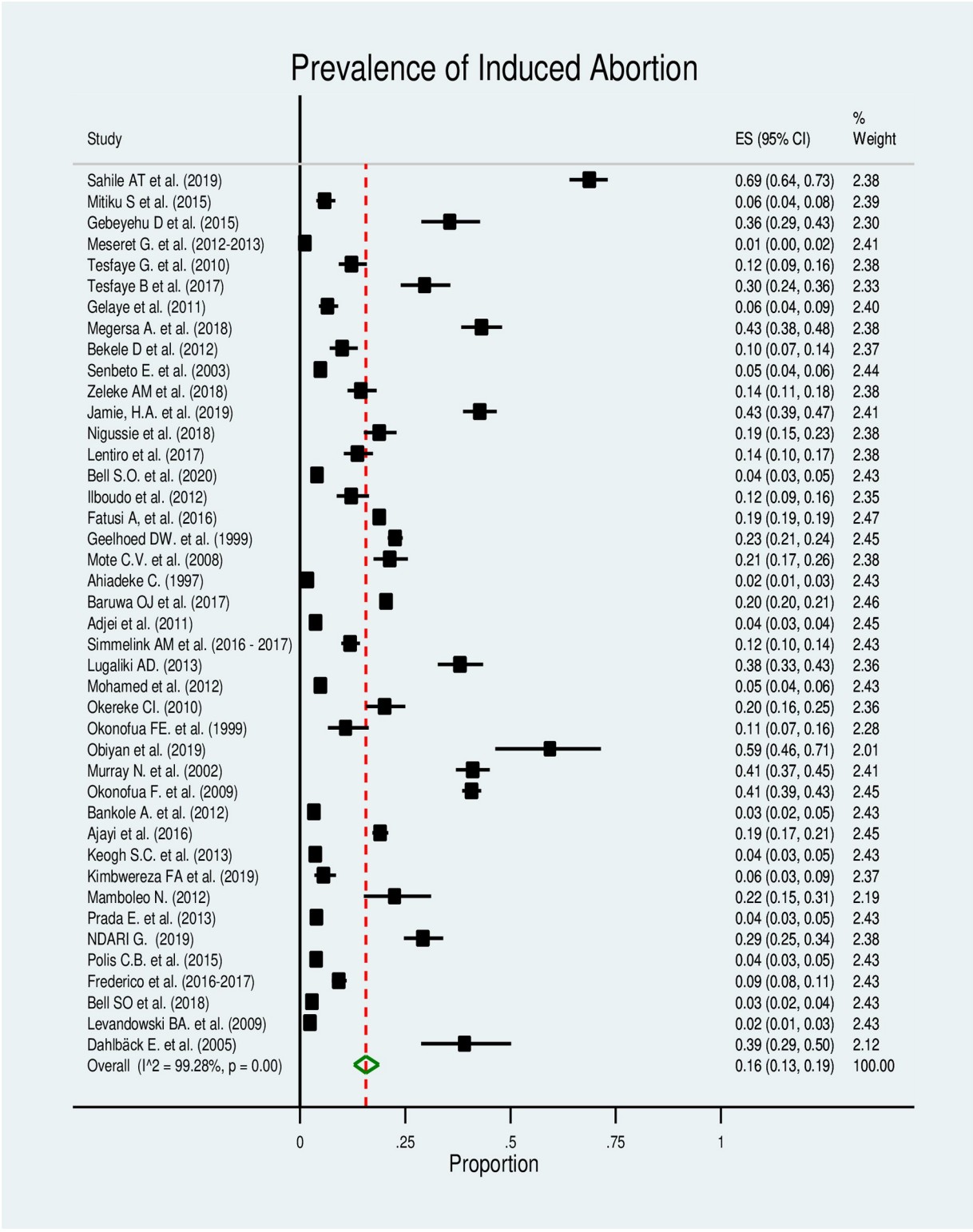

**Fig 2. Forest plot for the prevalence of the induced abortion in Africa.**

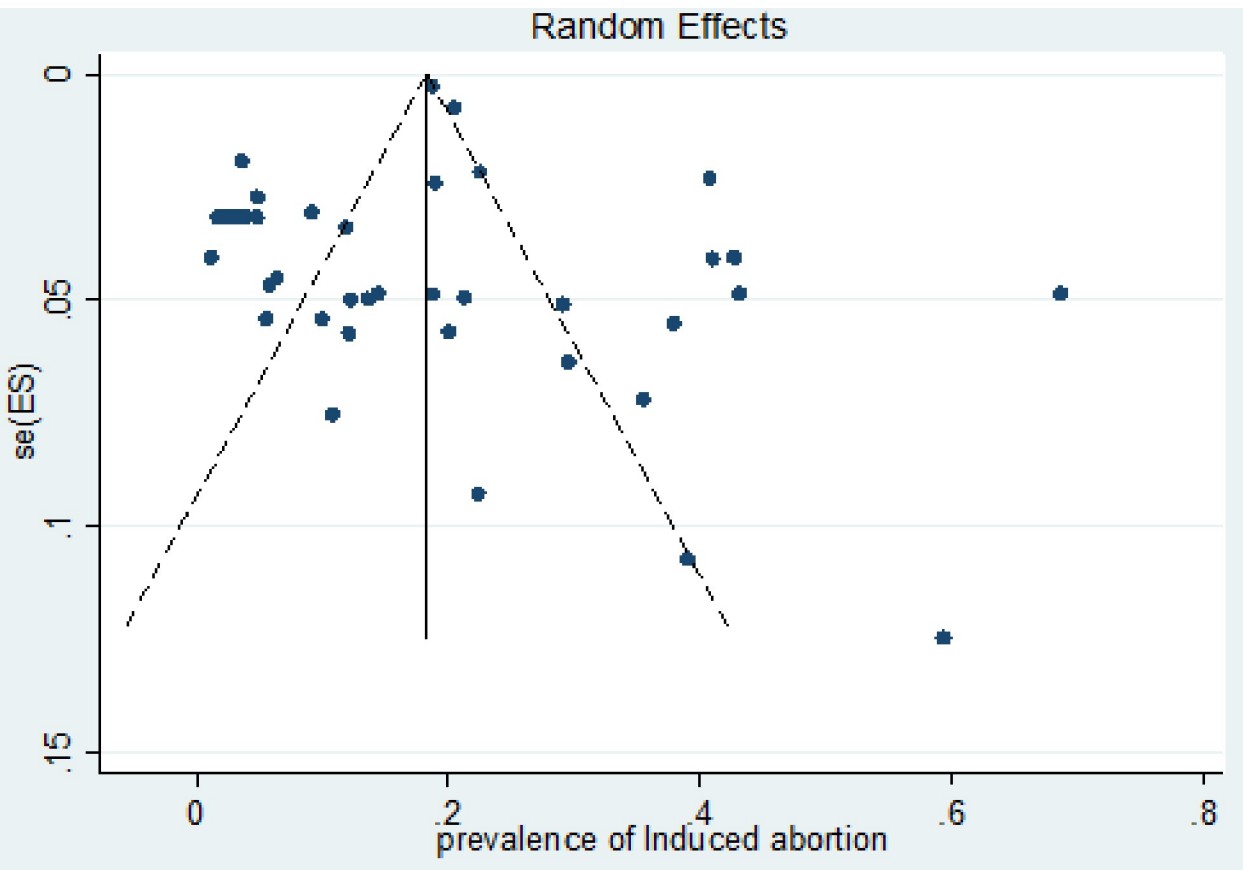

**Fig 3. Funnel plot included distribution of studies in induced abortion.**

this subgroup analysis was to compare these overall estimates across groups and determine whether the considered grouping helps us explain some of the observed between-study heterogeneity. However, the subgroup analysis done does not change the heterogeneity observed ($I^2 > 99\%$, $p \leq 0.000$). Further, a meta-regression was run considering the sample size and study year as covariates to check if they were the sources of heterogeneity for the pooled prevalence of induced abortion, but none of them were also significant.

In the sub-group analysis by country (Fig 4), most studies were conducted in Ethiopia, and the pooled prevalence of induced abortion was 19% (95% CI: 10%–30%). Similarly, the subgroup analysis by year of study showed that the prevalence of induced abortion was 39% (95% CI: 17%–64%) among studies conducted in 2019 (Fig 5).

## Factors associated with induced abortion in Africa

The pooled effect estimates of associated factors, including unintended pregnancy (AOR = 9.51, 95% CI: 3.31–27.34), being an unmarried woman (AOR = 4.49, 95% CI: 2.46–8.20), educational status (AOR = 2.63, 95% CI: 1.7–4.06), and substance use (AOR = 2.72, 95% CI: 1.38–5.34), were significant predictors of induced abortion in Africa with a considerable level of heterogeneity ($I^2 = 76.5\%$, $p \leq 0.005$), ($I^2 = 61.9\%$, $p \leq 0.022$), ($I^2 = 51.9\%$, $p \leq 0.052$) and ($I^2 = 52.3\%$, $p \leq 0.099$) respectively.

**Unintended pregnancy and induced abortion.** This type of meta-analysis comprised four studies [26, 33, 35, 40]. According to the pooled meta-regression analysis, unintended

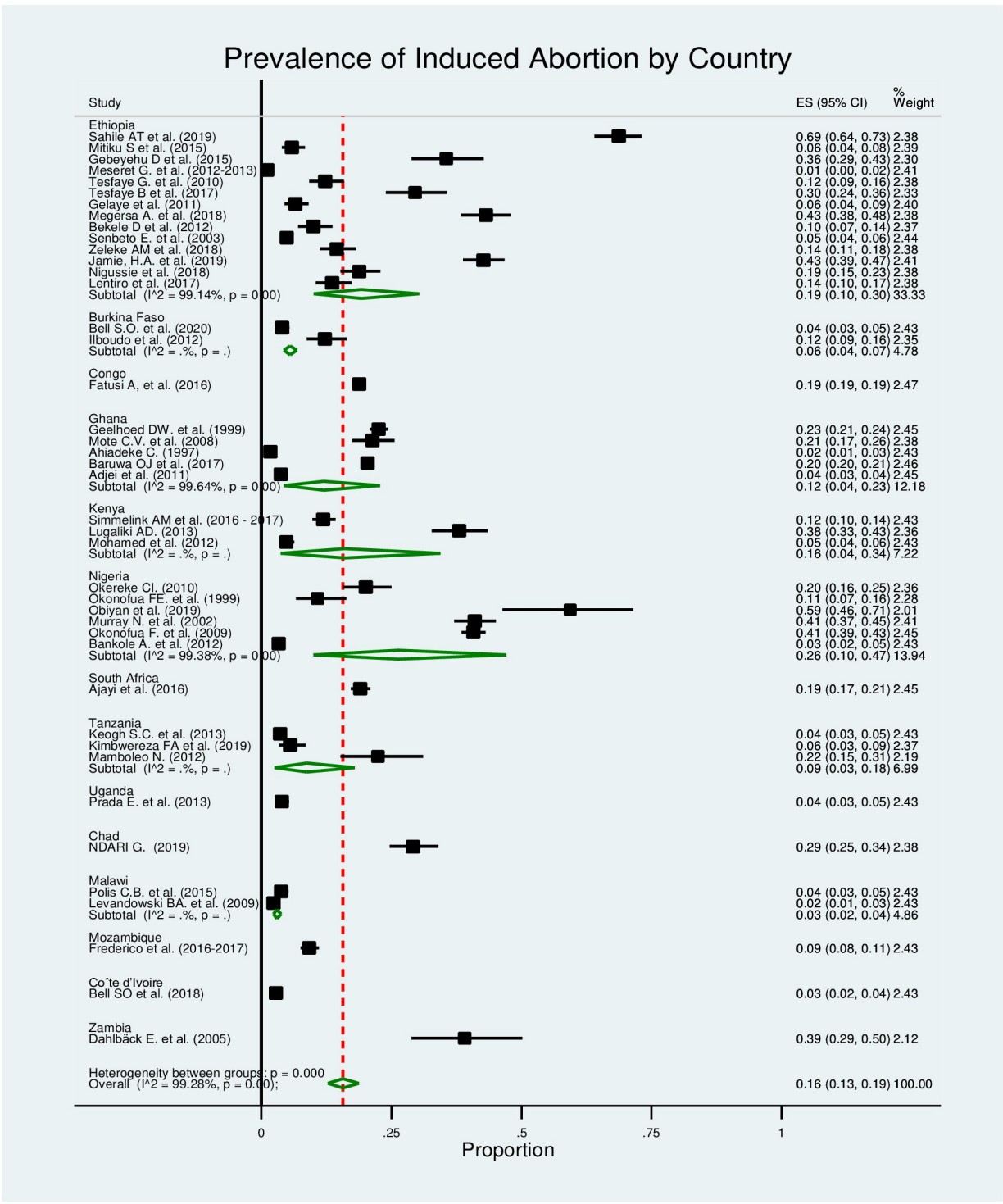

**Fig 4. Forest plot for the prevalence of induced abortion in Africa by country.**

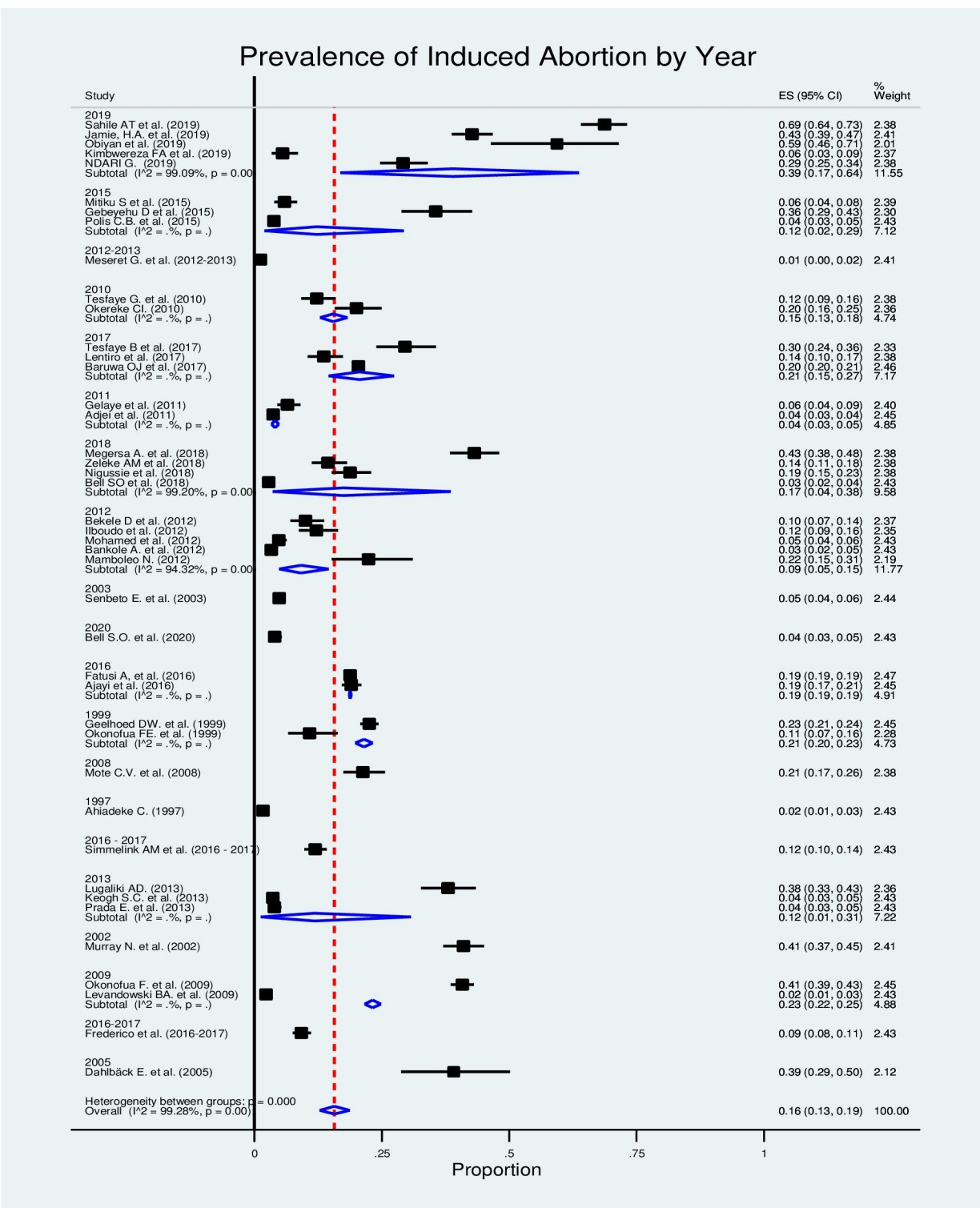

**Fig 5. Forest plot for the prevalence of induced abortion in Africa by year.**

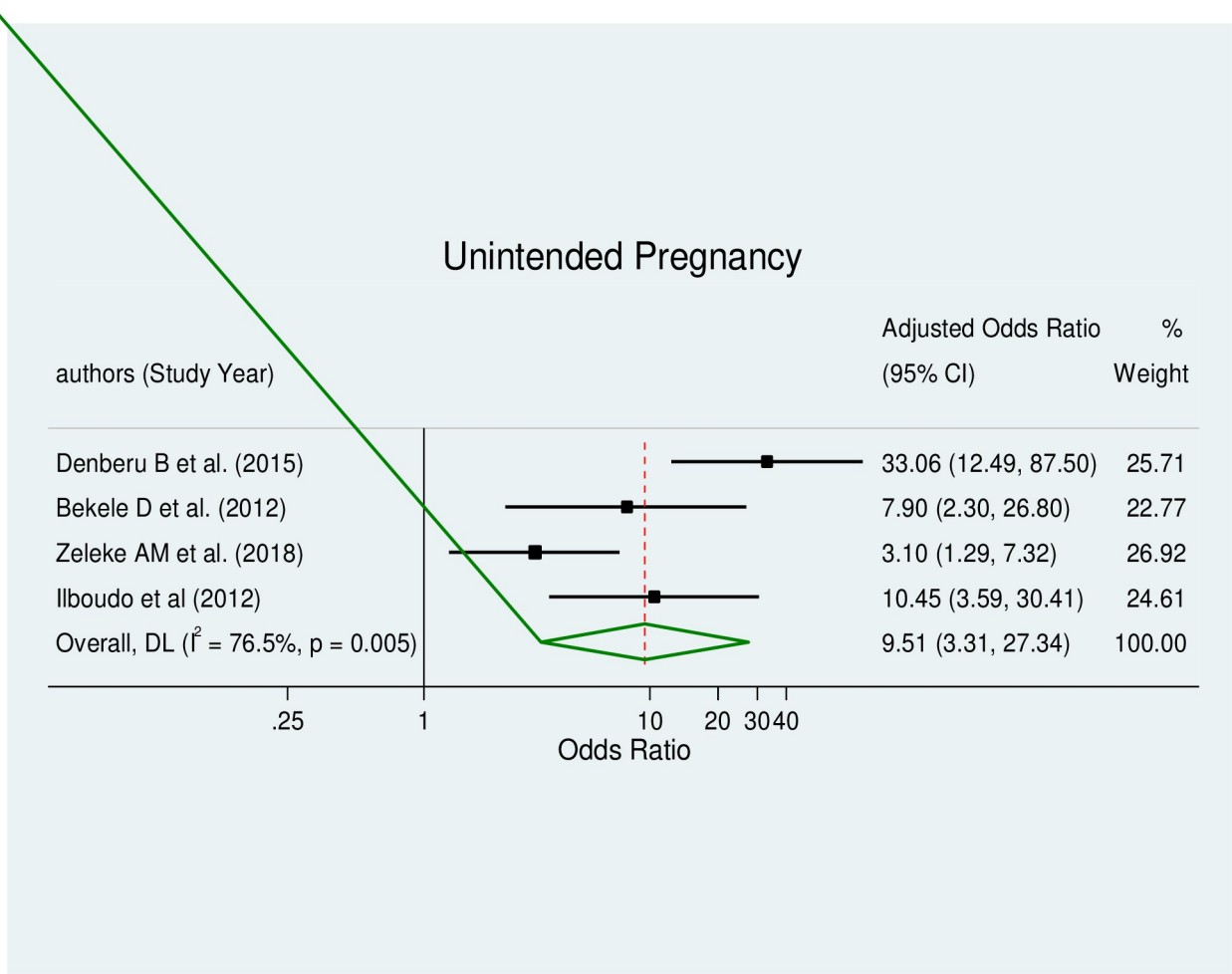

**Fig 6. Forest plot of unintended pregnancy and induced abortion.**

pregnancy had a statistically significant correlation with induced abortion (AOR = 9.51, 95% CI: 3.31–27.34) (Fig 6). This category of the meta-analysis showed high heterogeneity; hence, a random effect model was produced. Additionally, no evidence of publication bias was found using Egger's tests, which had a p-value of 0.958.

**Unmarried women and induced abortion.** The correlation between being an unmarried woman and having an induced abortion was evaluated in this review. Unmarried women were positively related to induced abortion (AOR = 4.49, 95% CI: 2.46–8.20), according to the results of a meta-regression analysis that included data from seven studies [28, 30, 36, 42, 44, 46, 48] (Fig 7). This category of the meta-analysis showed moderate heterogeneity. The test for publication bias using Egger's test revealed no publication bias with a p-value of 0.347.

**Educational status and induced abortion.** Women with primary and secondary education had a considerably higher risk of having an induced abortion, according to the results of a pooled meta-regression analysis of eight studies [23, 28, 31, 32, 36, 44–46] (AOR = 2.63, 95% CI: 1.7–4.06) (Fig 8). There was some moderate heterogeneity across the included studies. The results of Egger's test for the small study effect and the existence of publication bias were both significant (p = 0.000). Therefore, a trim-and-fill analysis for the educational status was performed in order to lessen and correct the study's apparent publication bias. The outcome

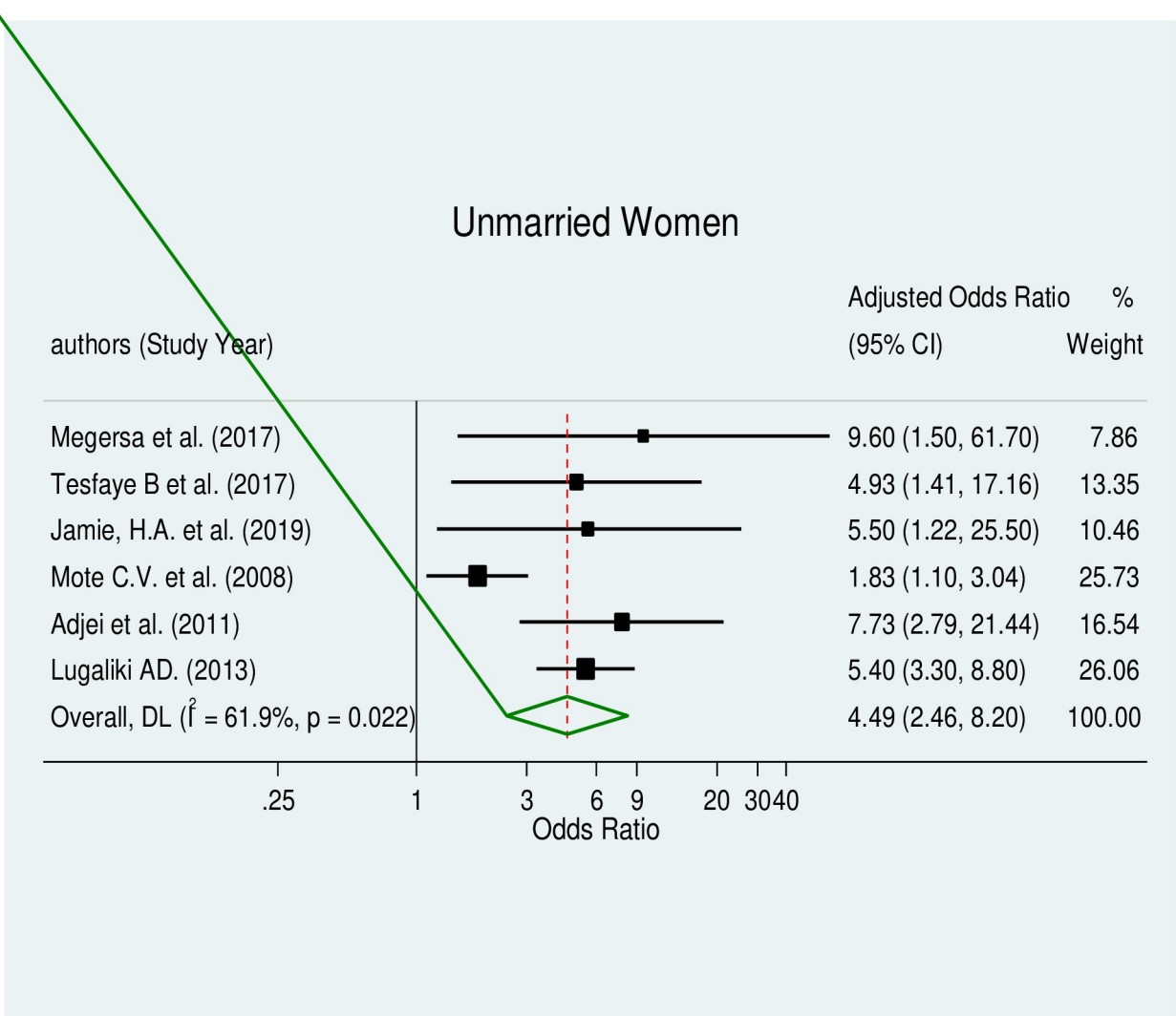

**Fig 7. Forest plot of unmarried women and induced abortion.**

indicated that two studies had now been imputed in place of the missing studies and that, following correction, the estimated pooled odds ratio had marginally changed.

## Substance use and induced abortion

Women who used drugs or alcohol were significantly more likely to have an induced abortion (AOR = 2.72, 95% CI: 1.38–5.34) (Fig 9). The four included studies [24, 31, 38, 55] showed moderate heterogeneity, and Egger's tests for the small study effect revealed a significant small study effect and the presence of publication bias (p = 0.029). As a result, a trim-and-fill analysis for substance usage was carried out in order to lessen and correct the research's apparent publication bias. No studies were imputed for missing studies, according to the outcome.

## Discussion

Induced abortion continues to be a major concern for women's reproductive health and human rights around the world. To determine the prevalence of induced abortion among

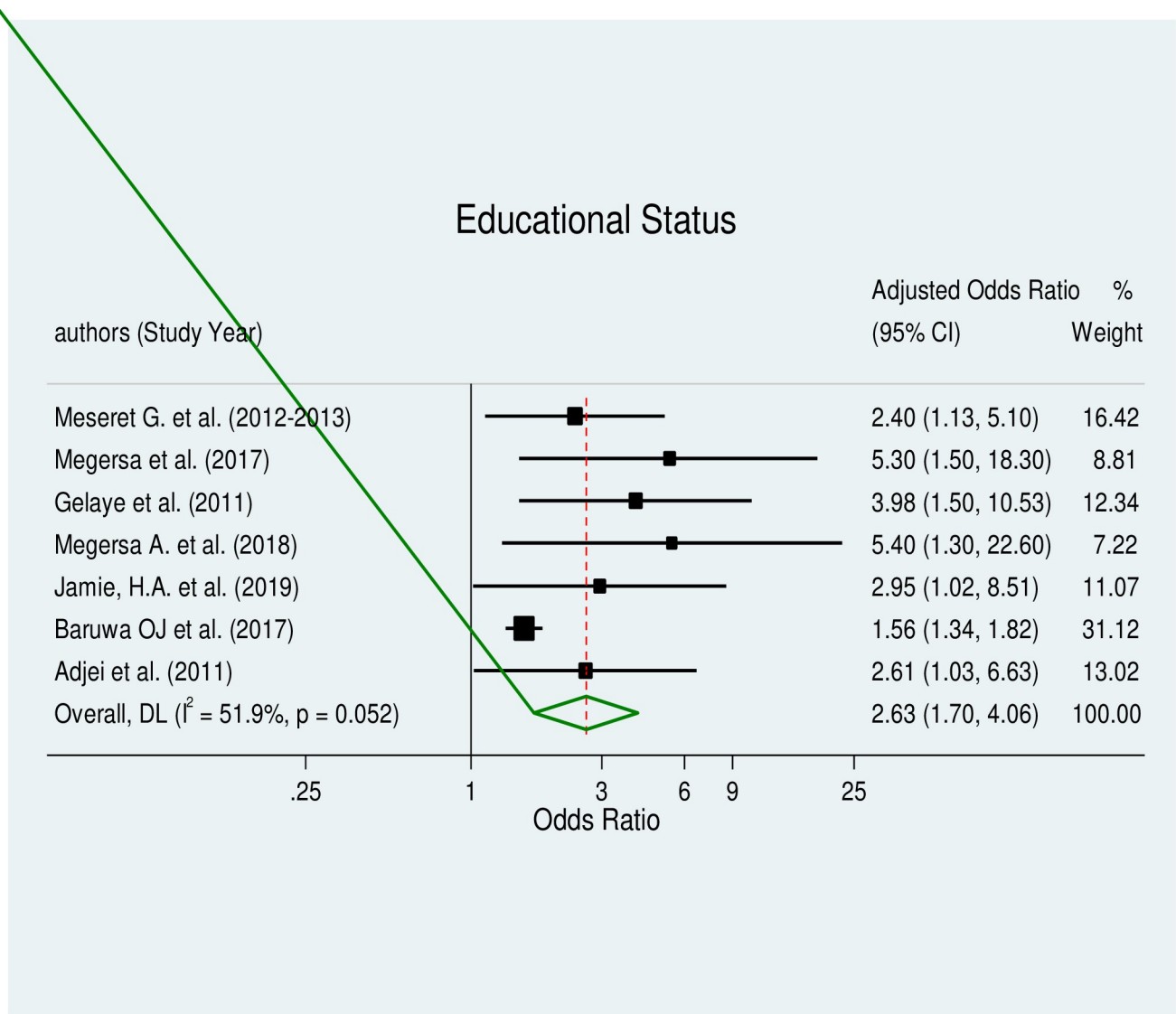

**Fig 8. Forest plot of educational status and induced abortion.**

reproductive age groups in Africa, a systematic review and meta-analysis were undertaken. According to this study in Africa, the pooled prevalence of induced abortion among reproductive age groups is 16% (95% CI: 13%-19%), or 160 per 1000 reproductive age groups of women.

The result of this study is much higher than the studies carried out in the Globe and Ethiopia with the same study design, which revealed 5.81% and 5.06% [66, 67] respectively. Similarly, studies conducted in East Africa, worldwide in 2010–14, Sierra Leone, Indonesia, Iran, Turkey, Ethiopia, and Australia revealed that 7.79%, 35 abortions occurred annually per 1000 women, 9%, 42.5 abortions per 1,000 women, 3.8%, 10.9%, 23 per 1,000 women and 2.1/100 women [68–75] respectively. The difference might be due to the difference in study design, sample size, cultural, social, religious norms related to abortion, and access to reproductive health service including family planning and safe abortion.

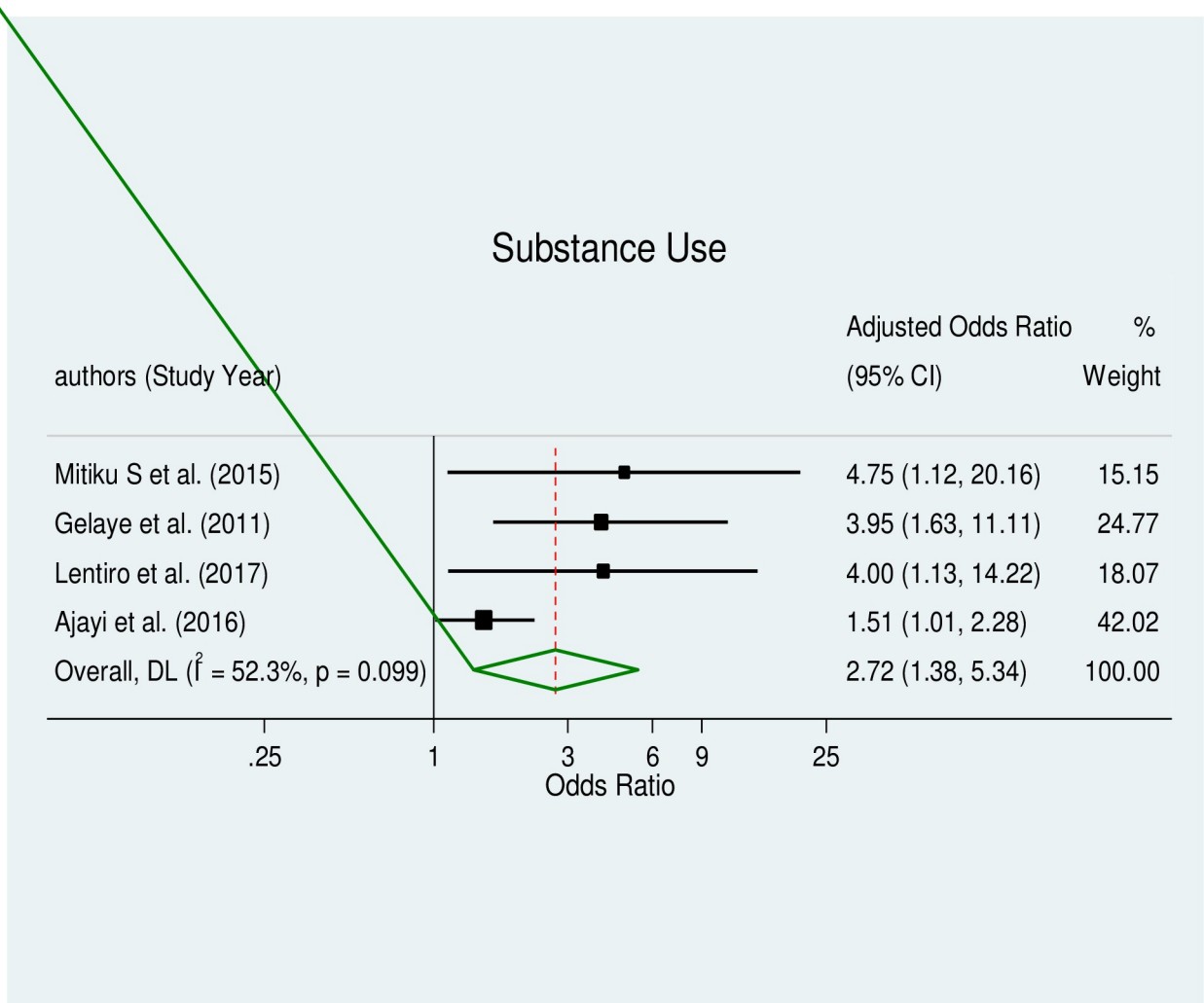

**Fig 9. Forest plot of substance use and induced abortion.**

A study conducted in China from in different study year, and Nepal showed that 24.0%, 28.95%, and 21.1% [76–78] respectively, indicated a higher rate of induced abortion than our finding. The possible explanation for the discrepancy might be due to the difference in study design, sample size, data collection methods, type of respondents, and the general public's level of education. On the other hand, a study was done in China 16.70% [79] and our pooled prevalence (16%), which is what we found, more coincide.

Our study found a strong association between unintended at the time of pregnancy and induced abortion, with women who had unintended at the time of pregnancy being nearly ten times more likely to undergo induced abortion compared to those with intended at the time of pregnancy. This finding is consistent with previous studies from around the world, including Turkey, Iran, and India [6, 73, 80, 81], which have also demonstrated a significant relationship between unintended at the time of pregnancy and induced abortion. These results highlight the importance of addressing unintended at the time of pregnancy through effective family planning programs and education to reduce the incidence of induced abortion and improve women's reproductive health.

In this study, unmarried women were 4.49 times more likely to develop induced abortion than their counter part. Studies conducted in East Africa, Sierra Leone, and China have reported contrary result [68, 70, 79]. The discrepancy could be the cultural and social norms surrounding marriage and sexuality in these different regions.

Our meta-regression analysis showed that women with primary and secondary education were 3 times more likely to undergo induced abortion than their counter parts. This finding is consistent with studies conducted in East Africa, Sierra Leone, Iran, and Nepal [68, 70, 72, 78]. However, there is a contrary finding from a study conducted in China [79], which reported a lower prevalence of induced abortion among women with higher educational status. The contrasting results could be due to contextual factors, such as cultural and social difference across regions and countries as well as difference in study design and data collection methods.

According to this study, women who used drugs or alcohol were more likely to have an induced abortion than those who did not. This result is consistent with a study conducted in Turkey [73]. This suggests that the association between drug and alcohol use and induced abortion is not specific to a particular region or country. The collective evidence from these studies highlights the importance of addressing substance use disorders and providing appropriate interventions to help reduce the incidence of induced abortion.

## Strengths and limitations the study

To the best of our knowledge, this is the first systematic review and meta-analysis of the prevalence of induced abortion and its associated factors in Africa. Therefore, data from this study are needed to support reproductive and adolescent health programmers, administrators, policymakers and to formulate recommendations for future clinical practice and guidelines. In addition to the aforementioned strengths, prior to conducting this manuscript, the protocol was not registered or published online.

## Conclusion

The pooled prevalence of this study is 16%. The results of this study thus imply that the pooled prevalence of induced abortion is higher than that of earlier studies that were published in some nations. Because of the social stigma, privacy issues, and worry about legal repercussions, abortion continues to be one of the most delicate sexual and reproductive practices. Induced abortion continues to be a serious public health issue for African women and is one of the main causes of maternal morbidity and mortality.

Any effort to prevent variables like unwanted pregnancy and substance use through educational and contraceptive interventions must focus on eliminating the need for induced abortion, and it is crucial for women of reproductive age to have access to the right options. Safe abortion services should be promptly and easily available to all women who become pregnant unintentionally. Therefore, the data from this study are needed to support reproductive and adolescent health programmers and policymakers and to formulate recommendations for future clinical practice and guidelines.

## Supporting information

**S1 Table. PRISMA 2020 Checklist.**
(DOCX)

**S2 Table. Newcastle-Ottawa Quality Assessment Scale.**
(DOCX)

## Acknowledgments

We would like to thank the authors of the included studies for their contributions to this systematic review and meta-analysis study.

## Author Contributions

**Conceptualization:** Teklehaimanot Gereziher Haile, Teklehaymanot Huluf Abraha, Gebreamlak Gebremedhn Gebremeskel, Kidane Zereabruk, Tesfay Hailu Welu, Teklit Grum, Negasi Asres.

**Data curation:** Teklehaimanot Gereziher Haile, Teklehaymanot Huluf Abraha, Gebreamlak Gebremedhn Gebremeskel, Kidane Zereabruk, Tesfay Hailu Welu, Teklit Grum, Negasi Asres.

**Formal analysis:** Teklehaimanot Gereziher Haile, Teklehaymanot Huluf Abraha, Gebreamlak Gebremedhn Gebremeskel, Kidane Zereabruk, Tesfay Hailu Welu, Teklit Grum, Negasi Asres.

**Investigation:** Teklehaimanot Gereziher Haile, Teklehaymanot Huluf Abraha, Gebreamlak Gebremedhn Gebremeskel, Kidane Zereabruk, Tesfay Hailu Welu, Teklit Grum, Negasi Asres.

**Methodology:** Teklehaimanot Gereziher Haile, Teklehaymanot Huluf Abraha, Gebreamlak Gebremedhn Gebremeskel, Kidane Zereabruk, Tesfay Hailu Welu, Teklit Grum, Negasi Asres.

**Project administration:** Teklehaimanot Gereziher Haile, Teklehaymanot Huluf Abraha, Gebreamlak Gebremedhn Gebremeskel, Kidane Zereabruk, Tesfay Hailu Welu, Teklit Grum, Negasi Asres.

**Resources:** Teklehaimanot Gereziher Haile, Teklehaymanot Huluf Abraha, Gebreamlak Gebremedhn Gebremeskel, Kidane Zereabruk, Tesfay Hailu Welu, Teklit Grum, Negasi Asres.

**Software:** Teklehaimanot Gereziher Haile, Teklehaymanot Huluf Abraha, Gebreamlak Gebremedhn Gebremeskel, Kidane Zereabruk, Tesfay Hailu Welu, Teklit Grum, Negasi Asres.

**Supervision:** Teklehaimanot Gereziher Haile, Teklehaymanot Huluf Abraha, Gebreamlak Gebremedhn Gebremeskel, Kidane Zereabruk, Tesfay Hailu Welu, Teklit Grum, Negasi Asres.

**Validation:** Teklehaimanot Gereziher Haile, Teklehaymanot Huluf Abraha, Gebreamlak Gebremedhn Gebremeskel, Kidane Zereabruk, Tesfay Hailu Welu, Teklit Grum, Negasi Asres.

**Visualization:** Teklehaimanot Gereziher Haile, Teklehaymanot Huluf Abraha, Gebreamlak Gebremedhn Gebremeskel, Kidane Zereabruk, Tesfay Hailu Welu, Teklit Grum, Negasi Asres.

**Writing – original draft:** Teklehaimanot Gereziher Haile, Teklehaymanot Huluf Abraha, Gebreamlak Gebremedhn Gebremeskel, Kidane Zereabruk, Tesfay Hailu Welu, Teklit Grum, Negasi Asres.

**Writing – review & editing:** Teklehaimanot Gereziher Haile, Teklehaymanot Huluf Abraha, Gebreamlak Gebremedhn Gebremeskel, Kidane Zereabruk, Tesfay Hailu Welu, Teklit Grum, Negasi Asres.

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
