## [Decision Letter · Decision Letter 0]

26 Jan 2024

PONE-D-23-21853

Induced Abortion in Africa: A Systematic Review and Meta-analysis

PLOS ONE

Dear Dr. Haile,

Thank you for submitting your manuscript to PLOS ONE. After careful consideration, we feel that it has merit but does not fully meet PLOS ONE’s publication criteria as it currently stands. Therefore, we invite you to submit a revised version of the manuscript that addresses the points raised during the review process.

The manuscript has been evaluated by two reviewers, and their comments are available below.

The reviewers have raised a number of minor concerns, specifically they feel that including more details in the methodology section would strengthen this manuscript.

Could you please carefully revise the manuscript to address all comments raised?

We look forward to receiving your revised manuscript.

Kind regards,

Johanna Pruller, Ph.D.

Associate Editor

PLOS ONE

Journal Requirements:

Reviewers' comments:

Reviewer's Responses to Questions

**Comments to the Author**

1. Is the manuscript technically sound, and do the data support the conclusions?

Reviewer #1: Yes

Reviewer #2: Yes

2. Has the statistical analysis been performed appropriately and rigorously? 

Reviewer #1: Yes

Reviewer #2: Yes

3. Have the authors made all data underlying the findings in their manuscript fully available?

Reviewer #1: Yes

Reviewer #2: Yes

4. Is the manuscript presented in an intelligible fashion and written in standard English?

Reviewer #1: Yes

Reviewer #2: Yes

5. Review Comments to the Author

Reviewer #1: 1. Is the manuscript technically sound, and do the data support the conclusions?

Yes, the manuscript is technically sound, as it follows a systematic review and meta-analysis method to estimate the pooled prevalence of induced abortion and its associated factors in Africa. The data support the conclusions, as they show that the prevalence of induced abortion is high and varies by country and year of study.

2. Has the statistical analysis been performed appropriately and rigorously?

Yes

The statistical analysis has been performed appropriately and rigorously, as the authors use a systematic review and meta-analysis method to combine the results of different studies on the prevalence of induced abortion and its associated factors in Africa. The authors use software called STATA 14, which is a widely used and reliable tool for statistical analysis. They also use funnel plot and Egger regression test to assess the potential publication bias, which is the tendency of published studies to report more favorable or significant results than the true effect. They perform sub-group analysis by country and year of study to explore the sources of heterogeneity and to compare the results across different settings and time periods. The authors follow the standard and rigorous procedures for conducting a meta-analysis and reporting their findings.

3. Have the authors made all data underlying the findings in their manuscript fully available?

Yes

4. Is the manuscript presented in an intelligible fashion and written in Standard English?

Yes, The manuscript is presented in an intelligible fashion and written in Standard English, as it follows a clear and logical structure of background, methods, results, and conclusion.

Reviewer #2: 01) The manuscript titled "Induced Abortion in Africa: A Systematic Review and Meta-analysis" provides a comprehensive investigation into the prevalence of induced abortion in Africa and associated factors. The manuscript is well-structured, offering a clear understanding of the research scope, methods, results, and implications. Also, the topic of induced abortion in Africa holds immense importance in the realms of public health, reproductive rights, and policy development. Here are some specific points for consideration and potential improvements:

02) In reference to line 86, the inclusion of unpublished articles is highlighted by the authors. It is crucial that the authors offer a clear explanation regarding the decision to incorporate unpublished articles in the review. Additionally, readers would benefit from a detailed description of the methodologies employed to assure the quality and reliability of information derived from these unpublished sources."

03) In reference to line 90, where the authors indicate that 'All articles published up to May 1, 2023, were included in this study,' it would enhance clarity for readers if the date range, specifying both the starting and ending dates, is explicitly mentioned.

04) From line 120 to 131, the inclusion and exclusion criteria are provided as a list. It is suggested to consider presenting these criteria in a tabular format to enhance readability and the overall structure of the manuscript.

05) In line 130, the authors mention the exclusion of manuscripts with methodological problems. It is important for the authors to provide details on the nature of these methodological problems, how they were identified, and who made the decisions regarding exclusions. This information is crucial for maintaining objectivity and transparency in the review process.

06) In line 165, the authors state that 'At first, 976 studies were found through a thorough search of electronic databases.' It would be beneficial for readers to know how many articles were found from each database. Consider presenting these details in the PRISMA flow chart (figure 01) for improved clarity.

07) In reference to line 177, where the authors note the inclusion of studies with small sample sizes, as low as 64, alongside the exclusion of studies with methodological problems, there arises a concern about the consideration of sampling and sample size in the analysis of the papers' methodological comprehensiveness. It would be helpful for the authors to provide clarification on how the adequacy of sample size was assessed and whether it was a factor in evaluating the overall methodological quality of the included studies.

08) The manuscript indicates that the final sample comprises 46 studies, yet in line 189, the authors highlight the inclusion of only 42 studies in the meta-analysis. To enhance transparency and clarity regarding the study selection process, it is imperative for the authors to provide detailed explanations on why the remaining four studies were excluded from the meta-analysis

09) "In lines 279 to 281, the paper asserts that 'Our study found a strong association between unintended pregnancy and induced abortion, with women who had unintended pregnancies being nearly ten times more likely to undergo induced abortion compared to those with intended pregnancies.' It is suggested to rephrase this statement as 'unintended at the time of pregnancy / conception' to enhance clarity."

6. PLOS authors have the option to publish the peer review history of their article (what does this mean?). If published, this will include your full peer review and any attached files.

Reviewer #1: **Yes: **Getnet Melaku

Reviewer #2: **Yes: **M. Suchira Suranga

---

## [Author Response · Author response to Decision Letter 0]

12 Feb 2024

Date: February 08, 2024

Subject: Response to editors, and reviewers for the revised manuscript submission.

Title: Induced abortion in Africa: a systematic review and meta-analysis; [PONE-D-23-21853]

Dear Editor and Reviewers,

We are grateful for your careful reading of our work and your insightful comments. We truly value the time and energy you invested in offering thoughtful feedback and recommendations. Every one of your comments has been thoroughly reviewed, and the appropriate changes have been implemented in response. The individual reviewer comments have been addressed by explaining the adjustments made and offering a thorough response to each comment. We have also considered the journal requirements and ensured that our paper conforms to all relevant guidelines and formatting requirements. We believe that these changes have greatly improved our study's quality, precision, and clarity. We are certain that the revised manuscript now successfully addresses the issues brought up throughout the review process and complies with the journal's requirements. We would want to thank you again for all of your helpful advice and suggestions during this process. We welcome any further comments or advice you may have.

Sincerely,

The authors

Reviewer #1: 

1. Is the manuscript technically sound, and do the data support the conclusions?

Yes, the manuscript is technically sound, as it follows a systematic review and meta-analysis method to estimate the pooled prevalence of induced abortion and its associated factors in Africa. The data support the conclusions, as they show that the prevalence of induced abortion is high and varies by country and year of study.

Response: We appreciate your compliments on our manuscript. We appreciate you acknowledging the thorough systematic review and meta-analysis we carried out to estimate the prevalence of induced abortion in Africa and take into consideration differences by country and study year. Your evaluation of our methodology and the evidence supported by the data is quite positive. Once again, thank you so much for your valuable feedback.

2. Has the statistical analysis been performed appropriately and rigorously?

Yes

The statistical analysis has been performed appropriately and rigorously, as the authors use a systematic review and meta-analysis method to combine the results of different studies on the prevalence of induced abortion and its associated factors in Africa. The authors use software called STATA 14, which is a widely used and reliable tool for statistical analysis. They also use funnel plot and Egger regression test to assess the potential publication bias, which is the tendency of published studies to report more favorable or significant results than the true effect. They perform sub-group analysis by country and year of study to explore the sources of heterogeneity and to compare the results across different settings and time periods. The authors follow the standard and rigorous procedures for conducting a meta-analysis and reporting their findings.

Response: Thank you for taking the time to review and positive feedback. Your positive feedback on the rigor of our statistical procedures is greatly appreciated.

3. Have the authors made all data underlying the findings in their manuscript fully available?

Yes

Response: Thank you for your review and for recognizing that we have made all the data underlying the findings in our manuscript fully available. Once again, we appreciate your acknowledgement of our efforts in this regard.

4. Is the manuscript presented in an intelligible fashion and written in Standard English?

Yes, the manuscript is presented in an intelligible fashion and written in Standard English, as it follows a clear and logical structure of background, methods, results, and conclusion.

Response: Thank you for your review and positive feedback. Your recognition of our efforts in presenting the study clearly and using appropriate language is encouraging. Once again, thank you so much for your valuable assessment.

Reviewer #2: 

1) The manuscript titled "Induced Abortion in Africa: A Systematic Review and Meta-analysis" provides a comprehensive investigation into the prevalence of induced abortion in Africa and associated factors. The manuscript is well-structured, offering a clear understanding of the research scope, methods, results, and implications. Also, the topic of induced abortion in Africa holds immense importance in the realms of public health, reproductive rights, and policy development. Here are some specific points for consideration and potential improvements:

Response: We appreciate your compliments on our manuscript. Thank you for acknowledging the thoroughness of our research on the prevalence of induced abortion in Africa and associated factors. We are pleased to hear that the manuscript's structure effectively conveys the research scope, methods, results, and implications. Once again, we are very grateful for your review.

2) In reference to line 86, the inclusion of unpublished articles is highlighted by the authors. It is crucial that the authors offer a clear explanation regarding the decision to incorporate unpublished articles in the review. Additionally, readers would benefit from a detailed description of the methodologies employed to assure the quality and reliability of information derived from these unpublished sources."

Response: Thank you for taking the time to review and insightful comments regarding the unpublished papers we included in our manuscript. We recognize that unpublished research plays a significant role in offering insightful information and increasing the sample size and number of studies included in the analysis, which can improve the statistical power and precision of the findings. We provide a thorough explanation of the procedures followed to evaluate the accuracy and consistency of data obtained from unpublished sources, including the application of the Newcastle-Ottawa Quality Assessment Scale. We appreciate you bringing these crucial points to our attention once more. We value your feedback, and we hope that this will provide a clear explanation and a detailed description regarding the inclusion of unpublished articles.

3) In reference to line 90, where the authors indicate that 'All articles published up to May 1, 2023, were included in this study,' it would enhance clarity for readers if the date range, specifying both the starting and ending dates, is explicitly mentioned.

Response: We appreciate your review and insightful suggestion. We believe that providing readers with precise information about the starting and ending dates will enhance clarity. Therefore, based on your suggestion, we have added a clear and explicit date range in our amended manuscript (lines 89-90), specifying both the starting and ending dates. We are grateful for bringing this to our attention and for your suggestions on improving the clarity of our study.

4) From line 120 to 131, the inclusion and exclusion criteria are provided as a list. It is suggested to consider presenting these criteria in a tabular format to enhance readability and the overall structure of the manuscript.

Response: We appreciate your evaluation and insightful recommendation. We agree that organizing these requirements in tabular format can make our study easier to read and better organized overall. In our revised version (line number 119), we have included a table to enhance the understanding of the inclusion and exclusion criteria. Once again, we appreciate you bringing this to our notice and helping to improve the clarity and presentation of our study.

5) In line 130, the authors mention the exclusion of manuscripts with methodological problems. It is important for the authors to provide details on the nature of these methodological problems, how they were identified, and who made the decisions regarding exclusions. This information is crucial for maintaining objectivity and transparency in the review process.

Response: We appreciate your review and valuable feedback regarding the need for additional information on the nature of methodological problems, how they are identified, and the decision-making process regarding exclusions. In our amended manuscript (Table 1, line 120), we have included more precise information on the nature of methodological problems encountered in the studies. We also identified these methodological problems through a careful review of the studies to ensure accuracy and consistency in the evaluation process. We appreciate your feedback, and we will ensure that the revised manuscript provides greater clarity on the nature of methodological problems, their identification, and the decision-making process for exclusions. Thank you for bringing these important aspects to our attention.

6) In line 165, the authors state that 'At first, 976 studies were found through a thorough search of electronic databases.' It would be beneficial for readers to know how many articles were found from each database. Consider presenting these details in the PRISMA flow chart (figure 01) for improved clarity.

Response: Thank you so much for your review and insightful feedback. We have updated the PRISMA flow chart (Fig 1) in our revised manuscript to reflect the number of articles that were found in each database. We have taken note of your advice and made the appropriate changes to our manuscript. We appreciate you pointing this out to us and helping to make our manuscript better.

7) In reference to line 177, where the authors note the inclusion of studies with small sample sizes, as low as 64, alongside the exclusion of studies with methodological problems, there arises a concern about the consideration of sampling and sample size in the analysis of the papers' methodological comprehensiveness. It would be helpful for the authors to provide clarification on how the adequacy of sample size was assessed and whether it was a factor in evaluating the overall methodological quality of the included studies.

Response: Thank you very much for your review and valuable feedback. In our manuscript (lines 175-178), we have provided an explanation of how we assessed the studies included in our analysis. While inadequate sample size is considered an indication of a methodological problem for inclusion in the analysis, the studies included in our study met the criteria of the Newcastle-Ottawa Quality Assessment Scale, which evaluates the quality and accuracy of the study. Furthermore, it is crucial to remember that study quality and eligibility for inclusion in a systematic review and meta-analysis are not solely determined by sample size. In addition, other elements including study design, methodology, and bias risk should be considered. We appreciate your feedback on the importance of sample size and its role in evaluating the overall quality of our study. Once again, we sincerely appreciate your contribution in bringing these concerns to our attention, and we value your efforts to improve the quality of our study.

8) The manuscript indicates that the final sample comprises 46 studies, yet in line 189, the authors highlight the inclusion of only 42 studies in the meta-analysis. To enhance transparency and clarity regarding the study selection process, it is imperative for the authors to provide detailed explanations on why the remaining four studies were excluded from the meta-analysis

Response: We appreciate you bringing up the discrepancy between the final 46 studies and the inclusion of just 42 studies in the meta-analysis in the manuscript. We apologize for any confusion caused by this inconsistency and appreciate the opportunity to provide clarification. In our revised manuscript (lines 163-168), we have provided a more explicit explanation to clarify this distinction and ensure transparency regarding the inclusion and exclusion of studies in the meta-analysis and associated factors analysis. We appreciate you bringing this to our notice and helping to make our study more transparent and clearer.

9) "In lines 279 to 281, the paper asserts that 'Our study found a strong association between unintended pregnancy and induced abortion, with women who had unintended pregnancies being nearly ten times more likely to undergo induced abortion compared to those with intended pregnancies.' It is suggested to rephrase this statement as 'unintended at the time of pregnancy / conception' to enhance clarity."

Response: We appreciate your evaluation and thoughtful advice regarding the phrasing of the statement about the association between unintended pregnancy and induced abortion. In line with your suggestion, we have rephrased the statement in the amended manuscript (lines 272-278) to clarify that the pregnancies were unintended at the time of pregnancy, in accordance with your advice. This modification will enhance the clarity of the statement and provide a more accurate representation of our findings. Thank you for bringing this to our attention, and we value your contribution to improving the clarity of our study.

---

## [Decision Letter · Decision Letter 1]

27 Mar 2024

PONE-D-23-21853R1Induced abortion in Africa: a systematic review and meta-analysisPLOS ONE

Dear Dr. Haile,

Thank you for submitting your manuscript to PLOS ONE. After careful consideration, we feel that it has merit but does not fully meet PLOS ONE’s publication criteria as it currently stands. Therefore, we invite you to submit a revised version of the manuscript that addresses the points raised during the review process.

We look forward to receiving your revised manuscript.

Kind regards,

Renato Teixeira Souza

Academic Editor

PLOS ONE

Journal Requirements:

Additional Editor Comments:

Thank you for the opportunity to manage the submission of this current manuscript. I had the opportunity to review the current and latest versions and the comments from the reviewers. The manuscript is improved and it is very well-written. In order to improve clarity and reproducibility, I suggest making the data extraction process and objectives related to the subgroup analyses done in the meta-analysis clear: unintended pregnancy, unmarried women, educational status, and substance abuse. It was not clear enough whether such analysis was planned before conducting the systematic review or whether it was conducted after retrieving papers and looking at the available data. Data-driven analysis can introduce many biases. Make it clear whether the study protocol has been previously registered or published.

Reviewers' comments:

Reviewer's Responses to Questions

**Comments to the Author**

1. If the authors have adequately addressed your comments raised in a previous round of review and you feel that this manuscript is now acceptable for publication, you may indicate that here to bypass the “Comments to the Author” section, enter your conflict of interest statement in the “Confidential to Editor” section, and submit your "Accept" recommendation.

Reviewer #1: (No Response)

Reviewer #2: All comments have been addressed

2. Is the manuscript technically sound, and do the data support the conclusions?

Reviewer #1: Yes

Reviewer #2: Yes

3. Has the statistical analysis been performed appropriately and rigorously? 

Reviewer #1: Yes

Reviewer #2: Yes

4. Have the authors made all data underlying the findings in their manuscript fully available?

Reviewer #1: Yes

Reviewer #2: Yes

5. Is the manuscript presented in an intelligible fashion and written in standard English?

Reviewer #1: Yes

Reviewer #2: Yes

6. Review Comments to the Author

Reviewer #1: (No Response)

Reviewer #2: The author's revised version exhibits significant improvements in various aspects. With minor editorial and language corrections, the manuscript is poised for acceptance and publication. We extend our best wishes for your future research endeavors and studies.

7. PLOS authors have the option to publish the peer review history of their article (what does this mean?). If published, this will include your full peer review and any attached files.

Reviewer #1: **Yes: **Getnet Melaku

Reviewer #2: **Yes: **M. Suchira Suranga

---

## [Author Response · Author response to Decision Letter 1]

6 Apr 2024

Dear Editor and Reviewers,

We would like to thank the editor and reviewers for their time and expertise in reviewing our manuscript. your valuable input has undoubtedly contributed to the enhancement of our manuscript. We appreciate your time and effort in reviewing the previous version and considering our responses to your comments. We are glad that the revisions we made in response to your previous comments have addressed your concerns and met your expectations. We have carefully considered and incorporated your suggestions into the manuscript to improve its quality, clarity, and scientific rigor. Once again, we sincerely appreciate your positive response and we are grateful for your thorough evaluation of our manuscript.

Additional Editor Comments:

Thank you for the opportunity to manage the submission of this current manuscript. I had the opportunity to review the current and latest versions and the comments from the reviewers. The manuscript is improved and it is very well-written. In order to improve clarity and reproducibility, I suggest making the data extraction process and objectives related to the subgroup analyses done in the meta-analysis clear: unintended pregnancy, unmarried women, educational status, and substance abuse. It was not clear enough whether such analysis was planned before conducting the systematic review or whether it was conducted after retrieving papers and looking at the available data. Data-driven analysis can introduce many biases. Make it clear whether the study protocol has been previously registered or published.

Response: We appreciate your careful reading of our manuscript and your insightful comments. We acknowledge the concerns expressed about the data extraction process and objectives related to the subgroup analyses, publication biases, and study protocol, as well as your recommendations for enhancing clarity and reproducibility.

Regarding the data extraction process, it is stated explicitly on pages 5-6 (lines 106–112) that "The data were separately extracted by three authors using a structured method of data collection." the objective of subgroup analyses is to compare these overall estimates across groups and determine whether the considered grouping helps us explain some of the observed between-study heterogeneity, and the studies were grouped based on the study area and study year, and an overall effect size was computed for each group, This is covered in detail on page number 12 (lines 204–215). "Unintended pregnancy, unmarried women, educational status, and substance abuse" are not subgroup analyses; rather, they are characteristics that significantly predict the likelihood of an induced abortion.

Regarding publication bias, we have reported the results of the funnel plot, Egger's test, and trim and fill test, all of which showed no significant publication bias; this suggests that the results of our meta-analysis are not unduly influenced by the selective publication of studies based on their outcomes. Page number 11-12 (lines 189-202).

We regret to notify you, nonetheless, that a study protocol was not registered or published for this research. We recognize that in order to improve transparency and reduce biases, pre-registration or publication of the study protocol is crucial. We have taken many precautions to reduce potential biases by adhering to established guidelines for systematic reviews and meta-analyses, following a rigorous methodology, and providing a detailed description of our methodology and analysis approach, even though the lack of a registered or published protocol is a limitation. Finally, the lack of a registered or published protocol is mentioned as a limitation for this study, on page 16 (lines 298–303) of our work. We hope to keep things transparent and let readers know about this restriction by doing this. Once again, we appreciate your thorough review and constructive feedback.

---

## [Decision Letter · Decision Letter 2]

15 Apr 2024

Induced abortion in Africa: a systematic review and meta-analysis

PONE-D-23-21853R2

Dear Dr. Haile,

We’re pleased to inform you that your manuscript has been judged scientifically suitable for publication and will be formally accepted for publication once it meets all outstanding technical requirements.

Kind regards,

Renato Teixeira Souza

Academic Editor

PLOS ONE

Additional Editor Comments (optional):

Reviewers' comments:

Reviewer's Responses to Questions

**Comments to the Author**

1. If the authors have adequately addressed your comments raised in a previous round of review and you feel that this manuscript is now acceptable for publication, you may indicate that here to bypass the “Comments to the Author” section, enter your conflict of interest statement in the “Confidential to Editor” section, and submit your "Accept" recommendation.

Reviewer #2: All comments have been addressed

2. Is the manuscript technically sound, and do the data support the conclusions?

Reviewer #2: Yes

3. Has the statistical analysis been performed appropriately and rigorously? 

Reviewer #2: Yes

4. Have the authors made all data underlying the findings in their manuscript fully available?

Reviewer #2: Yes

5. Is the manuscript presented in an intelligible fashion and written in standard English?

Reviewer #2: Yes

6. Review Comments to the Author

Reviewer #2: The authors have addressed all the recommendations of the editor. I have no further suggestions from my side.

7. PLOS authors have the option to publish the peer review history of their article (what does this mean?). If published, this will include your full peer review and any attached files.

Reviewer #2: **Yes: **M Suchira S Suranga

---

## [Editor Report · Acceptance letter]

26 Apr 2024

PONE-D-23-21853R2 

PLOS ONE

Dear Dr. Haile, 

I'm pleased to inform you that your manuscript has been deemed suitable for publication in PLOS ONE. Congratulations! Your manuscript is now being handed over to our production team.

Kind regards, 

on behalf of

Dr. Renato Teixeira Souza 

Academic Editor

PLOS ONE